# Examining the exposure-reception-retention link in realistic communication environments via VR and eye-tracking: The VR billboard paradigm

**Ralf Schmälzle** *, Sue Lim, Hee Jung Cho, Juncheng Wu, Gary Bente

Department of Communication, College of Communication Arts and Sciences, Michigan State University, East Lansing, Michigan, United States of America

* schmaelz@msu.edu

**Data Availability Statement:** Data and analysis scripts are available from the OSF repository (https://doi.org/10.17605/OSF.IO/HPGZU). Our laboratory data is available from the Github

## Abstract

Exposure is key to message effects. No effects can ensue if a health, political, or commercial message is not noticed. Yet, existing research in communication, advertising, and related disciplines often measures 'opportunities for exposure' at an aggregate level, whereas knowing whether recipients were 'actually exposed' to a message requires a micro-level approach. Micro-level research, on the other hand, focuses on message processing and retention, takes place under highly controlled laboratory conditions with forced message exposure, and largely ignores how recipients attend selectively to messages under more natural conditions. Eye-tracking enables us to assess actual exposure, but its previous applications were restricted to screen-based reading paradigms lacking ecological validity or field studies that suffer from limited experimental control. Our solution is to measure eye-tracking within an immersive VR environment that creates the message delivery and reception context. Specifically, we simulate a car ride down a highway alongside which billboards are placed. The VR headset (HP Omnicept Pro) provides an interactive 3D view of the environment and holds a seamlessly integrated binocular eye tracker that records the drivers' gaze and detects all fixations on the billboards. This allows us to quantify the nexus between exposure and reception rigorously, and to link our measures to subsequent memory, i.e., whether messages were remembered, forgotten, or not even encoded. An empirical study shows that incidental memory for messages differs based on participants' gaze behavior while passing the billboards. The study further shows how an experimental manipulation of attentional demands directly impacts drivers' gaze behavior and memory. We discuss the large potential of this paradigm to quantify exposure and message reception in realistic communication environments and the equally promising applications in new media contexts (e.g., the Metaverse).

repository (https://github.com/nomcomm/vr_ billboard_paradigm). These datasets contain the raw fixation data for all analyses and the accompanying analysis scripts allow others to fully reproduce all results.

**Funding:** The author(s) received no specific funding for this work.

**Competing interests:** The authors have declared that no competing interests exist.

# Introduction

Imagine driving down a seemingly never-ending highway, the billboard signs that line the road occasionally catching your attention. You briefly glimpse at some, examine others more closely, and completely bypass others. What will you remember when you reach your destination and why? Here we suggest that combining VR technology with eye-tracking makes studying this phenomenon experimentally tractable, enabling researchers to link rigorously quantified message exposure to outcomes.

The paper is organized as follows: First, we introduce the concept of exposure and discuss its theoretical relevance within communication, advertising, and related disciplines. We distinguish between traditional measures of "opportunities for exposure", which are typically estimated at aggregate levels, and "actual exposure", which requires a measurement approach at the micro level. Next, we discuss experimental research on incidental memory as the type of memory that matters most for research on message effects. We argue that research on the exposure-reception-retention nexus should be done in ecological conditions, which is a challenge for current methodologies. We then introduce the combination of eye-tracking and virtual reality (VR) as a promising solution. Finally, we present the current study in which people are driving down a (virtual) highway with billboards along the way and eye-tracking is used to measure whether or not they look at individual billboards. We examine how exposure, now quantified objectively via eye-tracking, relates to subsequent memory and how this is influenced by a manipulation of participants' attention.

## Background

**Exposure as the theoretical cornerstone of message effects.** Exposure is key to media and message effects [1–4]. It is obvious that if audiences are not exposed to a message, communication can not have any effect—just like a pill not swallowed cannot have any pharmacological effect. This applies to all kinds of messages—from advertisements in newspapers and on the web, commercials on TV or radio, or billboards along the road. Given the central theoretical role of exposure as a prerequisite of message effects, virtually all media track metrics like audience size or reach [5, 6], and there is solid evidence that these relate to outcomes like brand awareness or ad recall [1, 2, 7–9]. However, strictly speaking, such approaches only quantify "opportunities for exposure." To keep within the pill analogy, this is like knowing how many pills were delivered or bought by consumers (opportunity to be exposed), but not whether an individual also took the pill (actual exposure). Thus, there is theoretical ambiguity with respect to the precise meaning of exposure: The term exposure refers to the state when a *message makes contact with a recipient* [1–3]. In practice, however, this kind of exposure (i.e. the contact between the message and the recipient) is not actually measured, but rather assumed follow from messages being *made available* in peoples' information environments [1–6].

This theoretical ambiguity regarding the definition of exposure translates into methodological ambiguity regarding its measurement, leading to a divide between frequently used macro-level measures of "exposure" (as a shorthand for "exposure opportunities") and almost absent approaches for measuring exposure at the micro level. Most measures of exposure opportunities refer to aggregate measures of message availability in the information environment, such as the volume or frequency of messaging in various channels (e.g., how often a message is on TV and the average audience at the time, how many drivers pass by a billboard, etc.). However, it is important to be clear that exposure as a theoretical variable must be conceptualized at an individual, or micro-level, i.e. when a message enters from a person's information environment into their neurocognitive system.

To make it clear why these measures can diverge, consider the following situation: A driver passes by a few billboards along a highway. Traditional metrics used in this context refer to aggregate statistics, such as how many cars travel along this highway (possibly weighted by the average number of passengers, time of the day, etc.). However, these metrics leave it open whether a given individual actually looked at any of the billboards. The same applies to ads on TV. Aggregate metrics can give us estimates about how many millions are tuned in to a program, ad segment, and so forth. However, we do not know whether a given individual actually looked at a specific ad.

Zooming in on this nexus between exposure and reception is challenging though. Doing so requires a micro-level approach that is capable of determining whether a person actually notices and subsequently processes a message that is in their information environment. This must be done on an individual-by-individual and message-by-message basis, and it requires novel measures that go beyond macro-level association studies between opportunities of exposure and outcomes [5]. However, doing so is theoretically necessary because otherwise the information from media ratings remains detached from the neurocognitive reception mechanisms and the conceptual definition of exposure does not match its measurement, which undermines validity.

**Incidental memory as a primary effect of exposure.** Despite the theoretical ambiguity around exposure and the resulting macro-micro divide in its measurement, it is clear that a primary consequence of exposure is memory—the retention of messages in the mind [5]. Indeed, impacting memory is one of the main goals of most messaging efforts [8, 10].

The kind of memory that is most relevant to exposure research is incidental memory, which is memory formed without the intention to memorize [10–13]. This is the default state we are in when encountering media messages in our daily lives. Under such circumstances, attention is typically deployed to items that are intrinsically salient or relevant, and the resulting incidental memories are thus most relevant for message reception and effects research. More broadly, extensive research about memory has shown that there is a link between attention (or distraction) and subsequent memory [14, 15] and that the way in which participants encode items (e.g., superficially or more elaboratively; [16, 17] strongly affects the resulting incidental memories.

In sum, research on exposure and memory are theoretically related because the former (exposure) is presumed to be the cause and the latter (memory) the effect. However, just like the difficulties in measuring exposure have hindered theoretical integration, memory research incurred methodological obstacles as well: Most of our memories are formed while we are engaged in everyday activities, but much memory research focuses on more deliberative memory tasks in which participants are instructed to memorize certain items. Moreover, laboratory studies of memory tend to be done under artificial conditions and rely on "forced exposure" in which people are instructed to look at on-screen messages. While this approach has led to important insights into memory formation, critics have long demanded that memory be studied under ecological conditions that focus more on everyday memory and incidental memory for messages we encounter [18–22].

**Examining the exposure-reception-retention nexus under ecological conditions: Significance and obstacles.** As established above, exposure is the basis of message effects and thus fundamentally important for all theories of mass communication, advertising, and media effects. However, the causal link between exposure, reception, and retention remains understudied. This is largely due to two methodological obstacles: first, the challenge of measuring actual exposure at the individual level, and second, the challenge of conducting memory research in more naturalistic settings, especially for psychological memory research but also for work specifically focusing on memory for messages [10, 23–28]. These methodological

limitations hold significant theoretical implications as they hinder the valid measurement of the theorized phenomena (i.e., actual exposure) and pose threats to ecological validity (i.e., laboratory memory tasks that may not generalize to real-world memory for messages). Consequently, this situation impedes deeper mechanistic insights.

If we want to delve into the processes of information reception and retention, we must take a micro-level perspective and measure exposure as it occurs [9, 29]. Furthermore, both exposure and memory should be studied in a more incidental and ecological manner, which is more representative of how messages are encountered and processed. Yet such conditions are difficult to realize. As a result, we know a lot about processes that occur post-exposure [21, 30], but these steps occur after the initial act of looking at a message. In other words, our understanding of the sequence from media content to reception processes to subsequent effects suffers from a measurement gap at a very early stage. In the following sections, we will introduce two methods—eye-tracking and virtual reality (VR)—that hold promise to overcome these limitations and thus promote method-theory synergy in the study of exposure [86].

**Eye-tracking for measuring exposure: Strengths and current limitations.** Eye-tracking is a clear candidate for examining the exposure-reception nexus for visual messages. Eye-tracking provides direct information about where an individual is looking [31], which is closely related to what messages a person is paying attention to. Due to these characteristics, eye-tracking is widely used to study how individuals respond to messages, where and for how long they look, and so forth [32–36].

However, previous eye-tracking studies were mostly conducted in controlled settings, and most eye-tracking research in the context of message reception was done using screen displays. This approach is perfectly valid for studying how people browse websites, read online reviews, or engage with other related modes of messaging [36, 37], but it does not apply to other messaging environments, such as highways, streets, and other contexts in which we encounter messages that are not screen-based [38]. In summary, we know much about situations in which people are placed in front of screens to study how they attend to displayed messages but considerably less about unconstrained information environments in which exposure happens more incidentally. Said differently, eye-tracking has been used profitably to study *how* people look at on-screen messages, but less so *whether* they look at messages in more natural messaging contexts. The former is clearly very important to examine perception, attention, and so forth, whereas the latter focuses on a more basic question—whether the self-determined *actual* exposure happens at all.

Of note, new wearable eye-trackers allow researchers to study exposure "in the wild" [38–40]. While these approaches are very promising for field or experience-sampling studies on natural exposure [41], they suffer from the challenge of all field research, which is a lack of control over the situation. Thus, while wearable eye-tracking enables us to learn much descriptive information about exposure in the real world, its potential for causal manipulation is limited. As an alternative to field studies, VR has been suggested as a methodology that unites experimental control and ecological validity [42–45]. In the following section, we first discuss the characteristics of VR, followed by the rationale for using VR in message exposure studies.

**VR's virtues: Realism, control, and measurement capability.** Virtual Reality (VR) can create immersive experiences, allowing researchers to simulate real-world environments and study human behavior in a controlled and systematic manner. This potential of VR has been documented in various contexts, including clinical psychological research, communication and advertising, as well as memory and navigation research, to name but a few [46–49]. Key characteristics that recommend VR for research use include its realism, its opportunities for experimental control, and its potential to integrate measurement. We will next expand on each of these beneficial characteristics.

First, VR can create increasingly realistic environments—whether it is riding a rollercoaster, walking along a virtual plank, or driving down a highway. Thus, researchers can design virtual environments that mimic the essential appearance of a wide variety of human visual environments. Of course, current VR applications are not perfect depictions of reality, but VR has proven to be a useful tool for examining a wide array of neurocognitive mechanisms [42, 50].

Second, because VR environments are digital creations, every detail can experimentally controlled, largely independent of physics and public policy rules. This is obviously a great asset in terms of experimental control, which is widely seen as one of the most critical features to experimentally demonstrate the causality of theoretical variables (mechanism and intervention potential). For instance, if the goal is to study the effects of billboards on a highway, researchers can create a virtual highway model and place a virtual billboard along the roadside. Critically, in many situations, such control is difficult and expensive to achieve (e.g., permits, cost factors), sometimes even completely impossible.

Third, it is relatively easy to integrate behavioral measures into VR studies. Some important measures come as a byproduct of the use of VR headsets (such as movement and orientation) and hand controllers (such as data on pointing and gripping activities). Newer hardware developments also focus on incorporating biobehavioral measurement devices into VR, such as face-tracking to study emotional expressions or heart rate, EEG, and eye-tracking measures [51–53].

These characteristics of VR—realism, control, and integrated measurement—suggest it is an ideal candidate for research on the exposure-reception-retention link, especially if paired with eye-tracking. Indeed, many promising VR-related applications are proposed for related research purposes [45, 51, 54–57]. Perhaps most closely related is a VR experiment that used a racing game with billboard ads along the track [58]. Memory for the ads was tested subsequently and found to be influenced by whether the drivers were actively driving or just passively observing the race. However, because the study did not measure eye-tracking, it could not ascertain to what degree the billboards were actually looked at and how this related to memory, which is what the current study set out to achieve.

## The current study and hypotheses

This study examines how quantified exposure to messages in a realistic environment relates to incidental memory of the messages. Based on the discussion above, we leveraged VR to simulate a drive down a highway alongside which billboards are placed. Moreover, we combine this VR environment with eye-tracking to capture exposure as it occurs. To further demonstrate the research potential of this new paradigm, we instructed half of the participants to count trash placed alongside the highway. The other half was instructed to look freely while driving down the highway. It is well documented that such a parallel, attention-consuming task will distract participants [14, 15, 59] and should consequently lead to fewer gaze fixations on the billboard messages [60, 61]. This simple manipulation also helps to connect the new VR billboard paradigm, which is primarily concerned with objectively assessing message exposure and its consequences for incidental memory, with the broader literature on message processing in which comparable tasks were often used [21, 22, 62].

Our main question was whether looking at individual billboards would predict subsequent memory. The theoretical reasoning on the exposure-retention link predicts that a substantial portion of messages that were looked at (i.e., for which actual exposure occurred) should be committed to memory. In addition to this hypothesis, we expected to confirm basic effects of task competition, namely that the instruction to look out for trash should lead to fewer billboard fixations and lower memory performance compared to the free viewing condition.

Finally, we wanted to explore general patterns of participants' viewing behavior in this situation. To this end, we conducted additional data-driven analyses to identify behavior patterns that would predict memory outcomes.

## Methods

### Participants

Forty participants ($m_{age}$ = 25.6, $sd_{age}$ = 11.2; 18 female) were recruited from a study pool and via word of mouth. The local review board approved the study, all participants provided written informed consent to the protocol, and student participants received course credit. One additional participant whose glasses did not fit under the VR HMD was immediately replaced, resulting in a final sample of 40 participants.

The sample size was set *a priori* to 40 participants, which was chosen based on power considerations. For instance, Neisser (1964) reported that unattended targets in visual search tasks were not remembered at all, and work on incidental memory suggests that even the shallowest encoding operations build some memory [19]. Moreover, the viewing condition should strongly impact how much participants look at the billboards, leading to much more fixations in the free-viewing than in the trash-counting condition [63]. These aspects combined suggest that billboards that were looked at should be recalled markedly more often than those that were not. We determined that for an assumed large effect ($d$ = 1.2), a sample size of 16 per group would be sufficient for high power ($1-\beta$ = 0.95, $\alpha$ = 0.05) to detect a between-group difference in the number of recalled billboards. We rounded this number up to 20 per group.

### Materials and equipment

**VR highway environment and billboard signs.** We developed a virtual highway in which 20 billboards were placed along the road. The core highway model was downloaded from Sketchfab.com and consisted of a digitized 3D photogrammetric model by the Nevada DOT [64]. It featured a straight stretch of Highway 50 taken near Cold Springs with arid and desert-like vegetation alongside. We added a sunny, blue sky with a few clouds and empty soda cans (used as targets for the trash-counting task). The twenty virtual billboard signs were placed along the road using 3D-billboard model stands, and the billboard messages were assigned to each of the billboard stands in a programmatic fashion, randomized across participants. The billboards were placed along the roadside semi-randomly, with the constraint that the inter-billboard distance had to be large enough to prevent interference [65].

**Billboard messages.** We developed 20 visual billboard messages using templates from Canva.com. Half of the billboards were about health-related topics (e.g., drinking, vaping, smoking, marijuana, seatbelt use, and distracted driving), resembling typical public service announcements that characterize outdoor billboard advertising in the U.S.. The second half of the billboards were typical advertisements (e.g., retail, lawyer services, hotels, and restaurants/food). The billboard messages all featured basic imagery along with some text, and their design was deliberately kept relatively simple but still typical of the kinds of billboards present on U.S. highways (see Fig 1 and online repository at github.com/nomcomm/vr_billboard_paradigm).

**VR and eye-tracking.** We relied on the Vizard VR software to create the VR environment, run the study, and track user behavior, including eye-tracking measurements (Vizard, 7.0; [66]. The VR device was an HP Reverb G2 Omnicept that includes eye-tracking capabilities. Participants used the right VR controller to accelerate and drive forward along the virtual highway. Because the highway was perfectly straight, no steering was required.

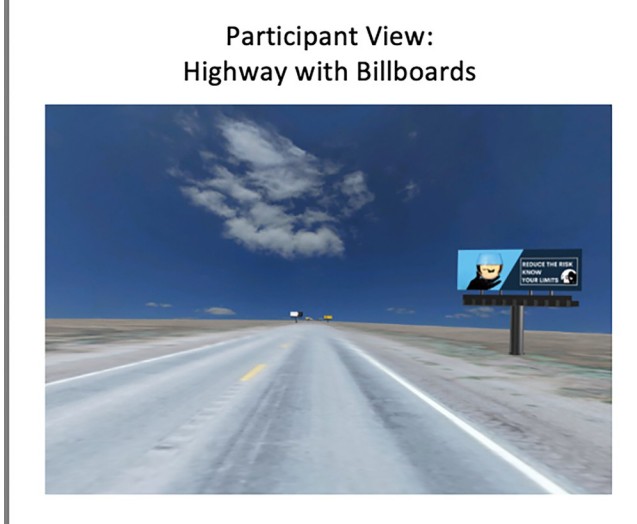

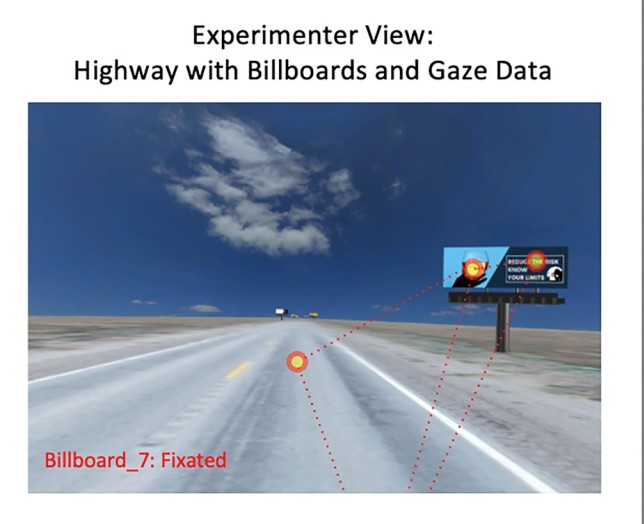

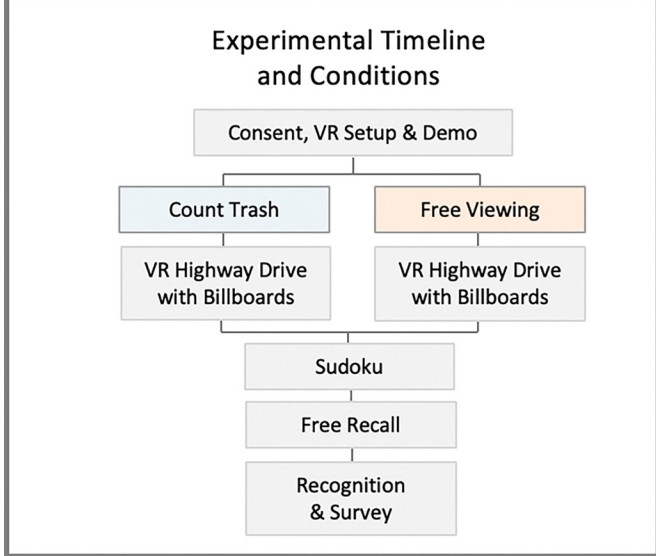

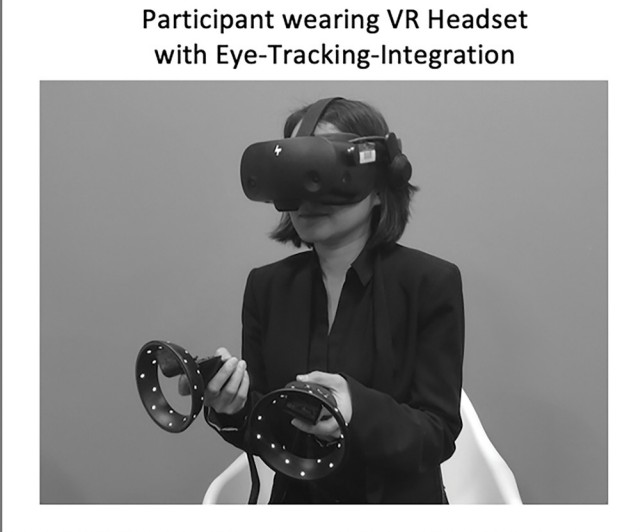

**Fig 1. Overview of the experimental protocol.** Top row: Screenshots of participant and experimenter views with one example billboard ('drunk_driving'). Bottom left: Experimental timeline and conditions. Half of the sample was instructed to count trash along the road; the other half was instructed to drive down the highway. Except for this difference in instruction, the highway and billboards were identical for both conditions. Bottom right: Participant wearing an HP Reverb G2 Omnicept VR headset.

## Experimental procedure and conditions

Once participants arrived and consented to the study, they completed a quick vision test, put on the VR headset, and underwent a calibration routine. Next, the participant entered a demo version of the study designed to acquaint them with VR, the virtual space, and the navigation. Then, the main experimental session was started, which involved driving down the virtual highway. Half of the participants were instructed to count the number of trash items in the environment (distraction condition). The other half were told to explore the environment while driving down the highway freely (free viewing condition).

After completing their virtual drive (which took about 10 minutes), participants were given a set of Sudoku puzzles for 2 minutes. Then, the experimenter conducted a structured

interview that asked participants about the number of trash items they saw, their general virtual driving experience, and which billboards they recalled (free recall task). As the last step of the study, participants completed an online questionnaire via Qualtrics that collected demographics, their experiences with the VR technology, and recognition of billboards. Specifically, we asked participants to report on their experience of spatial presence [67] and the occurrence of symptoms while in VR (VRSQ survey; [68]). For the recognition test, they were shown the 20 experimental messages and four distractors and asked whether they remembered seeing the messages during the highway drive. The purpose of the distractors was to gauge participants' tendency to recognize all messages as seen. Finally, participants were debriefed, and their eye-tracking data was saved.

**Main measures and analysis methods.** The main variable measured during the virtual drive was whether participants fixated on a given billboard. Fixations were detected algorithmically and saved to disk in a spreadsheet together with the fixated billboard's name. The fixation threshold was set to 0.25s. Thus, for every participant, the virtual drive yielded a spreadsheet containing where a given billboard was fixated (and how often, e.g., time 15s, fixation, billboard_1, etc.). Because billboard images were randomly assigned to individual billboard sign positions, a python program was written to resort the individual images to a given participant's eye-tracking data (e.g., time 15s, billboard_1, drunk_driving.jpg, . . .), allowing for subsequent data aggregation across participants and billboard messages.

The recall data (information on whether a participant brought up the billboard during the free recall task, e.g., "I recall seeing a billboard about drunk driving") was merged with the fixation information, and so was the recognition data (information on whether a participant recognized the billboard at the end of the study from a list of billboard images at the end of the study).

In the analysis, we first compared the number of fixations as well as the memory performance between conditions (free viewing vs. trash counting). These analyses were carried out to examine the overall effects of the condition (free viewing vs. trash counting) on viewing behavior (fixation count) as well as memory performance (measured via recall and recognition).

Next, the main analysis compared message recall between the experimental conditions, zooming in specifically on how viewing a billboard vs. ignoring it relates to memory and how this interacts with the instructed viewing condition. To this end, we used a mixed ANOVA with the factors *Fixation Status* (coding whether a message was looked at, 1 vs. not looked at, 0) and *Viewing Condition* (free viewing vs. trash counting). Note that the factor *Fixation Status* varies within subjects because every participant passed all 20 billboards and looked at some vs. ignored others. *Viewing Condition* represents a between subjects factor. In addition to computing this analysis using the recall data as the dependent variable, we also ran the same 2*2 ANOVA using the recognition memory as the dependent variable.

We document the analysis and provide code in the study's online repository at github.com/nomcomm/vr_billboard_paradigm.

## Results

We measured eye gaze while participants were cruising along a virtual highway flanked by billboards. One group could explore the environment freely while driving; another group was instructed to pay close attention to trash placed alongside the highway. Once they reached the end of the highway, we tested the participants' incidental memory for the billboard messages (they had not been told that they would be asked about the messages).

Participants' verbal comments about the study, collected during the verbal interview, revealed that they found the virtual highway drive realistic and engaging. The post-

experimental survey data confirms these observations. Specifically, participants reported experiencing spatial presence in the VR environment ($mean_{spatial\ presence}$ = 3.8; range 1–5, i.e., all items above the scale midpoint). Participants also reported almost no symptoms (e.g., dizziness, fatigue, or eyestrain; $mean_{VR-symptoms}$ = 1.36, range 1–4, i.e., all items below the scale midpoint).

## Fixations as a function of condition and general memory performance

To examine the effect of the experimental conditions (trash-counting vs. free-viewing), we compared the number of fixations to billboards. As predicted, we find that participants in the free-viewing condition had significantly more fixations on the billboards than participants in the trash-counting condition where participants' attention was directed more to the road than the billboards ($mean_{fixations:free-viewing}$ = 52.8, sd = 22.9; $mean_{fixations:trash-counting}$ = 21.8, sd = 16.1; $t_{38}$ = 4.98, p < 0.001; d = 1.58). These results are shown graphically in Fig 2 (left panel). Note that based on these numbers, it may be tempting to assume that all participants in the trash-count condition may have looked about once at every billboard, of which there were 20 in total. However, this was not the case. Rather, a few participants looked are some billboards more often, and many participants in the trash-counting condition did not look explicitly at many billboards.

Next, we examined the memory performance collected in the interview. In the free recall test, participants in the free-viewing condition recalled an average of 6.45 billboards (sd = 1.57, $recall\_rate_{free-viewing}$ = 0.32, significantly more than the average 2.95 (sd = 1.76, $recall\_rate_{trash-counting}$ = 0.15) billboards the participants in the trash-counting condition recalled ($t_{38}$ = 6.63; p < 0.001; d = 2.1; see Fig 2, middle panel).

Carrying out the same analysis on recognition data revealed even more pronounced results: Participants in the free-viewing condition recognized on average 14.6 billboards (sd = 3.25; $recognition\_rate_{free-viewing}$ = 0.73) compared to only 7.1 billboards recognized in the trash-counting condition (sd = 3.68; $recognition\_rate_{trash-counting}$ = 0.35), which is a highly significant difference ($t_{38}$ = 6.78, p < 0.001, d = 2.14, see Fig 2, right panel). Note that we do not report signal detection (SDT) analyses here because the upcoming analysis (see. Fig 3) does not allow directly computing SDT measures and we want to keep metrics consistent across the subsections of the results. However, the findings also hold when running the overall recognition analysis with dprime values instead of the recognition rate.

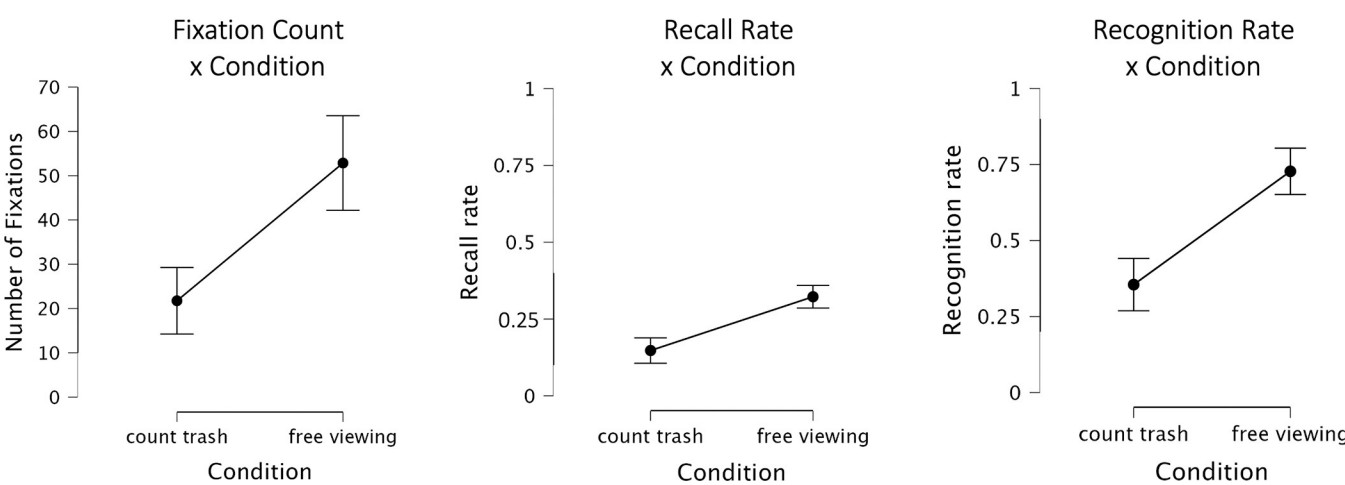

**Fig 2. Number of fixations, free recall, and recognition rates by condition.**

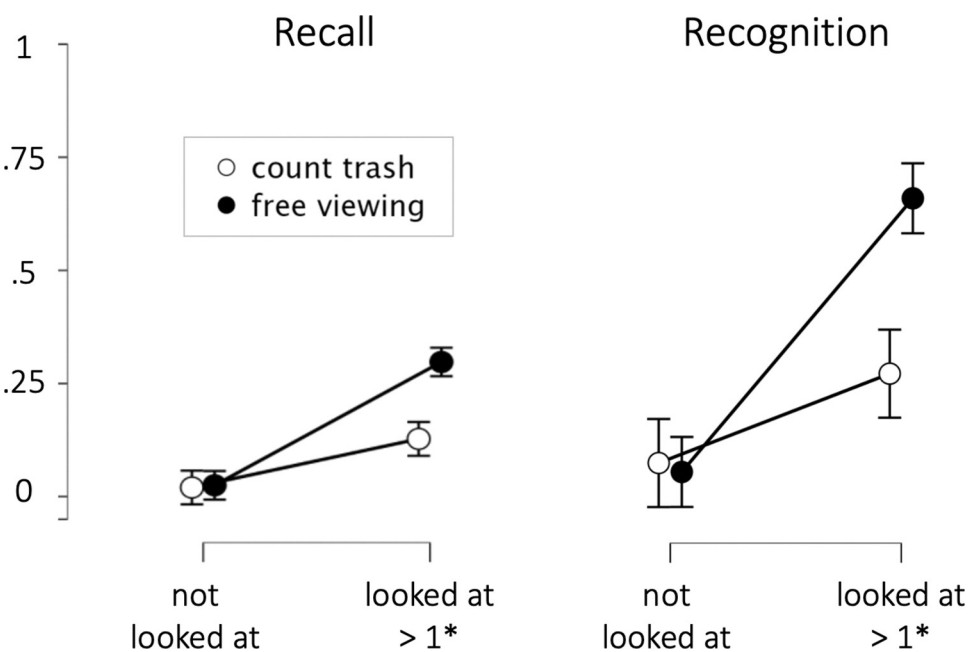

**Fig 3. Relationship between fixations and subsequent message memory.** Probability of recall (left) and recognition (right), based on whether a billboard was looked at (not fixated or fixated at least once) and condition (count trash vs. free viewing).

## Memory as a function of fixations status and viewing condition

Next, we focused on the relationship between fixations on individual billboards and subsequent memory for the billboards across the conditions. Toward this end, we determined for every participant the recall rate for looked-at billboards and the corresponding rate for not-looked-at-billboards ($recall\_rate_{looked-at-billboards}$ = (number of recalled billboards that were fixated at least once) / (number of billboards that were fixated at least once)). The resulting tabulation was combined with the information about the viewing condition and analyzed using ANOVA. Thus, the first factor (*Viewing Condition*: *free viewing vs. trash counting*) varied between-subjects (20 participants in the free-viewing and 20 participants in the trash-counting condition), and the second factor (*Fixation Status*: *looked-at vs. not-looked-at*) was a within-subject factor that was determined on an individual basis according to every participant's viewing behavior. For instance, one participant could have recalled 5 billboards out of 10 fixated ones ($recall\_rate_{looked-at-billboards}$ = 0.5) and 0 of the remaining 10 non-fixated ones, ($recall\_rate_{not-looked-at-billboards}$ = 0.0). By contrast, another participant could have recalled 8 out of 12 fixated ones and 1 out of the 8 non-fixated ones. The dependent variable in this 2*2 ANOVA was the free recall as our primary measure of memory performance. In addition to using free recall as the dependent variable, we also carried out the same 2*2-ANOVA for the recognition rate as another way of assessing subsequent memory.

This analysis revealed highly significant and consistent effects for both ways of assessing memory (recall and recognition rates): For free recall, there was a highly significant interaction effect between *Viewing Condition* and *Fixation Status* ($F_{(1,38)}$ = 25, $p < 0.001$, $\eta^2_p = 0.4$) and a highly significant main effect of *Fixation Status* ($F_{(1,38)}$ = 132.6, $p < 0.001$, $\eta^2_p = 0.78$). Follow-up tests confirmed higher recall in the free-viewing condition.

The results for the recognition data closely resembled the recall analysis: A highly significant main effect of *Fixation Status* ($F_{(1,38)}$ = 91.2, $p < 0.001$, $\eta^2_p = 0.71$) was qualified by a

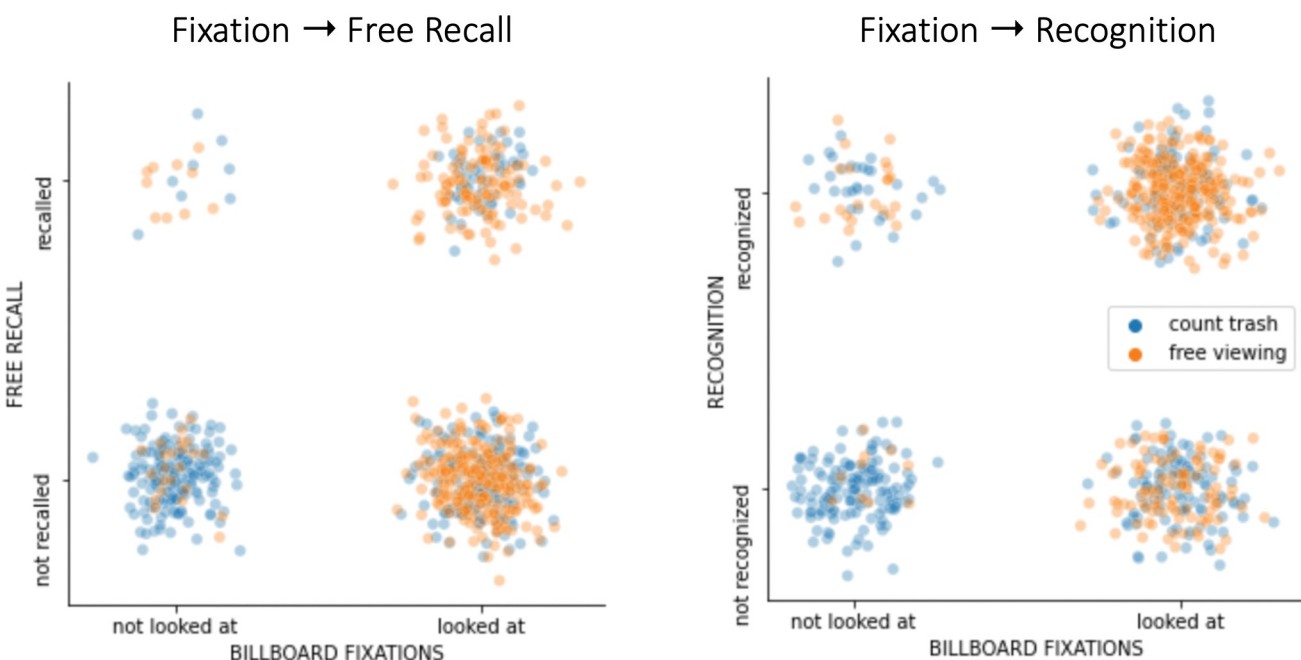

**Fig 4. Relationship between fixations and subsequent message memory at the level of single messages.** Left panel: Fixations and free recall performance. Every dot represents one billboard, color-coded based on whether participants were instructed to count trash (distraction) or view freely. Note that dichotomous variables (0—not looked at/not recalled, 1- looked at/recalled) were jittered randomly to aid visualization. Right panel: Same analysis but based on a recognition memory test.

significant ordinal interaction of *Viewing Condition* and *Fixation Status* ($F_{(1,38)}$ = 23.5, $p < 0.001$, $\eta^2_p = 0.38$). Follow-up tests again confirmed that recognition memory was higher in the free-viewing condition compared to the trash-counting condition. In other words, we find that if a billboard is looked at at least once, this boosts the likelihood it will be remembered by a factor of 5–20 (depending on the condition and how memory is measured).

To illustrate this more clearly, we created a plot that jointly visualizes whether a billboard was looked at and whether it was recalled or recognized, respectively, and in which condition (see Fig 3). As can be seen, in the trash-counting condition (blue dots), many billboards are not looked at. In the free-viewing condition (orange dots), more billboards are looked at (see results in the previous paragraph). Critically, however, the billboards that are never looked at are practically never brought up during the post-drive memory test (top left quadrant in the scatter plots, Fig 4; see discussion for explanation). The banners that were looked at (right column), are far more often recalled and almost always recognized.

To examine this strong contingency between looking and remembering at a more fine-grained level, we further unpacked the fixation data. Specifically, we extracted for every participant whether a billboard was never looked at, looked at a few times (i.e., at least once but less than that participant's medium fixation count across all 20 billboards), or looked at often (more than that participant's medium fixation count across all 20 billboards). The results of this analysis are illustrated in Fig 5, and they are statistically significant. A repeated measures ANOVA for the average number of items recalled (DV) revealed a strong effect of Viewing Behavior Intensity ($F_{(2,76)}$ = 12.1, $p < 0.001$, $\eta^2_p = 0.24$) with a significant interaction of viewing behavior intensity * viewing condition ($F_{(1,38)}$ = 3.68, $p < 0.05$, $\eta^2_p = 0.09$).

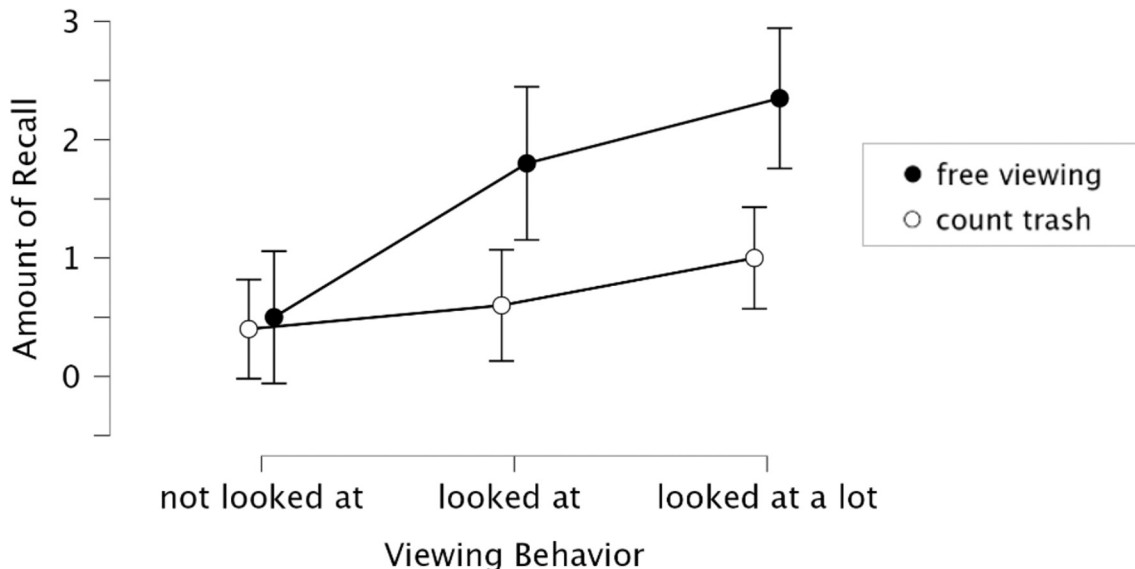

**Fig 5. Relationship between viewing behavior intensity and message recall at a more refined level (i.e., beyond looking vs. no-looking).**

### Exploratory analyses

The approach presented here affords predictive modeling. To this end, we used scikit-learn [69] to create a model that could classify whether a billboard would be recalled or not based on the existing variables, i.e., which billboard was presented (e.g., buckle-up, drunk_driving, hotel, etc.), how often the participant fixated it, in which position the item was viewed, and the condition (trash-counting vs. free-viewing). Using a 5-fold cross-validated SVC prediction, we found that this simple model performed well, with a ROC-AUC score of 72.8% compared to 50% for a dummy classifier (note that we used penalization to deal with the imbalance classes, i.e., recall being rarer than no recall; weighted F1 score: 0.58). Said differently, once we know that a participant looked at a given billboard, we can predict more accurately whether this participant will later recall it. This relationship can also be derived from the statistically significant effects and the data shown in Figs 3–5.

In addition to statistical testing and predictive models, we carried out additional analyses to examine false recognition, results for individual participants and individual billboards, effects of item position, and health vs. commercial billboard content.

To gauge the degree to which participants would be prone to false recognition, we included distractor billboards in the recognition set (i.e., billboards that were never seen). However, these distractors were only rarely falsely recognized, significantly less than all presented billboards, and only one participant misrecognized more than two distractor billboards. Thus, even though recognition measures can be prone to guessing, this is not the case here.

We also explored the relationship between fixations and memory and between different memory measures at the individual level. In the trash-counting condition, the number of fixations and memory measures were highly correlated (*r values* > 0.7, *p values* < 0.001), suggesting that participants who were more interested in the billboards or the study also remembered them better. In the free-viewing condition, this was not the case (*r values* were nominally even negative). In both conditions, however, recall and recognition were positively correlated (*r* = .54, *p* < 0.001 for the trash-counting condition, *r* = 0.2, *n.s.*, for the free-viewing condition). While these results are interesting and point to effects of motivation or interest, we opted not

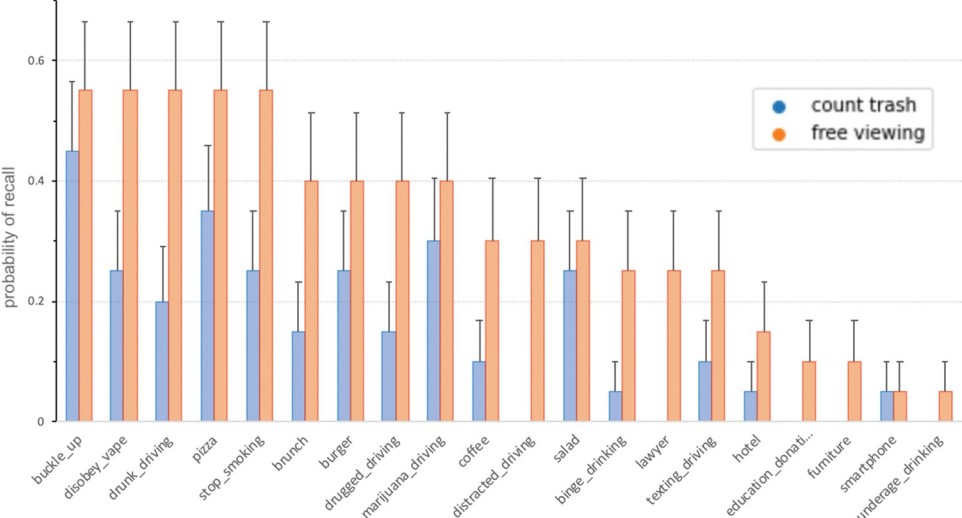

**Fig 6. Analysis for individual billboards.** Across both conditions (independent participants), the same billboards tended to be recalled more often.

to investigate them further because the current sample was relatively small for studying individual differences.

Moreover, we inspected the potential influence of the billboards' position (beginning vs. middle vs. end) on the probability of fixation, recall, or recognition. However, we did not find such effects nor evidence of an interaction with the condition. In both conditions, position curves were parallel and flat.

Inspection of the results for individual billboards revealed interesting effects: Specifically, as shown in Fig 6, some items were often recalled (e.g., buckle-up, disobey-vape, and burger) while others were barely remembered. This is also consistent with the predictive modeling result, where adding the item (one-hot-encoded) as a feature increased accuracy. Most likely, this is due to intrinsic differences between the billboards—either because of the topic's relevance to participants or because of low-level physical differences, such as saliency. Of note, we did quantify perceptual saliency [70] but did not see a significant relationship with memorability.

Lastly, we also explored the degree of memory for health-related vs. commercial billboards, finding no significant differences. Nominally, health-related billboards were slightly more often recalled (also see Fig 6), but the effect was insignificant ($F_{(1,36)} = 3.53$, $p = 0.07$). Across both conditions (independent participants), the same billboards tended to be recalled more often, as indicated by a significant vector correlation between trash-counting and free-viewing ($r = 0.82$, $p < 0.001$).

## Discussion

Messages are intended to inform and influence recipients. However, this requires that they are viewed, i.e., that audience members are actually exposed to the message. Therefore, exposure is a theoretical cornerstone of all message effects, but measuring exposure is challenging—especially at the individual level and within realistic messaging contexts. Here we created a VR paradigm that immerses users in an environment familiar to many: a drive down a highway with billboard signs along the road. Using a VR-integrated eye-tracker, we recorded whether participants looked at individual billboards, and we linked this information to subsequent memory

for the billboards. Our results show that this approach allows us to assess the exposure-reception-retention nexus rigorously, commensurate with how exposure is theoretically conceived, and in a realistic environment that boosts ecological validity and has good application potential.

## Discussion of main results

The current results are very clear and straightforward: The VR Billboard Paradigm enables studying whether people look at the messages they were exposed to. As simple as this sounds, the theoretical significance becomes apparent if one considers that exposure is often only inferred rather than actually measured. Clearly, these indirect ways of assessing exposure miss the point because what really matters for message effects is *actual reception* (i.e. the contact between message and recipients), not *fiat exposure* ("Let's hope people will look at the message"). Our paradigm makes it possible to study this and to do so in a way that strikes a balance between realism and experimental control.

The most important result is that participants' viewing behavior was significantly associated with message memory. Technically, one could have argued that all participants passed by all messages and thus had *opportunity for exposure* as conventionally theorized. However, measuring their visual information sampling via eye-tracking made it possible to measure actual exposure—and thus reception—and this explained whether billboards would be recalled. As such, the current paradigm enables following the arrow of causality—from exposure, to reception, to retention, in a seamless fashion, illustrating synergy between the innovative method and the theoretical framework [86].

Another key result was the strong influence of the attention-distracting task: Participants who were instructed to drive freely looked at the billboards more often and they recalled them more often. By contrast, participants who were instructed to count trash along the road showed very few fixations towards the billboards and generally low memory. Again, this perfectly matches our predictions that participants' attention would be consumed by the task, as it is well-known that attention and memory are tightly coupled [71–74].

These results all support our main argument, which is that exposure and reception are the prerequisites of message retention (memory). Specifically, the causal chain starts with the presence of a message in the information environment (opportunity for exposure), then the person noticing and taking in the message (actual exposure, reception), to subsequent memory (retention). Further evidence for this is also provided by the marked differences between fixations and memory in the trash-counting vs. free-viewing conditions and by the fact that messages that were looked at more were recalled more often.

We note that we are not the first to point to the potential of VR and eye-tracking for studying exposure and memory and that several related works exist. For instance, Kim et al. [75] have suggested a 360-degree video paradigm for measuring viewing behavior in naturalistic settings (e.g., 360-degree videos of real cityscapes). This approach combines realism and eye-tracking. Likewise, the work by Wang and Yao [58] is related to our approach (see introduction) and their findings regarding active playing vs. passive observation appear compatible with our finding regarding the influence of condition. From an experimental point of view, however, controlling billboards' placement and content or making message delivery contingent on behavior offers key strengths and innovations.

Going forward, we also expect key advances by integrating additional measures beyond the current eye-tracking. For instance, our results here focus on the eye gaze fixations and make hardly any use of pupil dilation or heart rate, both of which are already integrated into the HP Reverb G2 Omnicept headset. Kim et al. (2020), for instance, did combine their video with

MRI measurement. Although VR is challenging to combine with MRI (because the equipment is not compatible with brain scanners and head motion presents problems for MRI), other options exist and will likely become more widespread. These include EEG and fNIRS, which can provide additional insights into, e.g., the neural basis of memory formation and attention [73, 76, 77]. We also note that there were very few messages that were not looked at, but were still remembered (very rarely recalled freely, but sometimes recognized, see Fig 4). This can be explained by parafoveal or ultra-fast vision (i.e., below fixation threshold, [78–80]) and one could argue that these events are rare. Still, in such a case, neural measurements could add information beyond eye-tracking alone.

## Broader implications

The approach presented here holds significant value for understanding exposure and reception as the theoretical junction between message and receiver in communication. Although exposure has remained hard to study naturalistically in communication science, experimental memory research is an area in which exposure has always been manipulated—by forcing participants to attend to messages and then study the effects. As such, laboratory work on memory encoding and work flowing from incidental and ecological memory perspectives is complementary to the current approach [74, 81]. However, our emphasis differs by taking a communication perspective [1].

For instance, in memory research, the levels of processing framework highlighted how recall of memoranda varies based on processing depth [16, 82]. The core assumption is that the mental operations carried out over items (e.g., whether they are processed semantically or only superficially) influence the probability of recall. Such work has also found its way into communication science, for instance, via the popular elaboration-likelihood model and related work [83, 84]. Likewise, involvement in advertising has been proposed to refer to the degree of personal connections message recipients make with a message once they receive it [85]. Finally, the notion of exposure states also points to the importance of examining the psychological processes message recipients engage in once they are exposed to messages [29]. Thus, these different models and theories all have in common that they require measuring i) whether messages are received and ii) how people engage with them. The VR billboard paradigm presented here can definitely ascertain the former (whether messages are seen). To the extent that fixation amount and length can give insight into the latter (how messages are engaged with), we can also examine this with the current paradigm. Moreover, the paradigm can easily be expanded to measures like pupil dilation (or derivative metrics like fixation length, paths, etc.). In sum, the VR billboard paradigm resolves a longstanding theoretical problem in a new way [86].

Beyond these theoretical considerations, this approach clearly has significant applied potential as well: First, VR is heralded as the communication medium of the future, i.e., as an emerging media channel rather than just a methodologically advantageous gimmick [46, 47, 87, 88]. Indeed, the rebranding of the social media company FaceBook to Meta and mergers and acquisitions in the VR sector suggest that VR might become a messaging environment (or advertising space) soon. In other words, if people spend time in VR, they will be exposed to various messages while inside VR. Just like social media metrics (e.g., likes, comments, page impressions, and viewable impressions) have enabled the quantitative study of message diffusion on social networks, the rise of VR as a new messaging channel will thus create opportunities to connect data about quantified individual-level exposure (e.g., fixation to a message) to subsequent outcomes.

Second, the VR billboard paradigm directly applies to billboard advertising in the real world. For instance, it could be immediately used to empirically examine the effects of new

constructions on existing billboards (e.g., as legal testimony), forecast billboard effectiveness, and so forth. Thus, this approach can easily be adapted to other applied messaging questions because many message delivery contexts could be implemented in an equivalent manner. These include all forms of outdoor advertising, including airports, public transportation, and public spaces like Times Square in New York, the strip in Las Vegas, or any place where large audiences pass by. In all these cases, the ability to experimentally manipulate key characteristics of the appearance or the users' state and assess the effects of such manipulations on quantified user behavior (here, eye-tracking) could be of major value.

## Strengths, limitations, and avenues for future research

Key strengths of the VR billboard paradigm include that it is simple, realistic, flexible, and scalable. Using VR combined with eye-tracking to study message reception is not confined to billboards on highways but could be applied to other settings. It would be very simple to exchange the environment and use the same available Python code to detect fixations on messages in, e.g., city settings.

The biggest advantage of this approach over existing work (either screen-based eye-tracking or eye-tracking field research) is that it allows for the controlled testing of causal mechanisms while preserving a relatively high degree of realism. The ability to measure precisely and objectively and control variables experimentally are the key prerequisites for causal mechanism identification in e.g., the biological and behavioral sciences. These features are difficult to achieve in the social sciences, which often rely on macro-level association data. In this sense, the current paradigm holds great potential to overcome many limitations that have plagued message exposure research. Of note, though not the main focus of our study, this paradigm would seem equally promising for applied memory research [74, 81].

Like all research, the current study has several limitations. One limitation is that although the VR experience featured a version of a real highway drive (a digital twin of Highway 50 near Cold Springs), some elements of real life were missing (e.g., opposing traffic, birds, curves, and passengers through towns, etc.). Although driving on relatively empty highways is not uncommon, it is clear that adding these variables would make the simulation more realistic and generalizable. Likewise, our experimental messages are also limited in variety, number, or design- and content elements. For instance, we deliberately designed our billboard messages to feature no-name brands (e.g., lawyers, hotels, and nearby restaurants) and health-related public service announcements because these make up a large share of billboard advertising. We deliberately made these choices to balance experimental control and realism, but it could be argued that specific features matter. Fortunately, it is easy to add and test such factors, and high-realism driving games demonstrate that this is feasible (e.g., the popular Need for Speed or GTA series). For instance, future studies could add animated billboards, known brands, more controversial or evocative topics (like abortion-related billboards, ads for churches and casinos), or switch from the empty-highway driving situation to a city-like VR environment. Of course, there would also be the possibility of using mobile eye-tracking tools during a highway drive to conduct a field study [89].

Another limitation concerns the mostly student sample and its size. While our sample was adequate for the study's goal to demonstrate the value of this new paradigm by eliciting a fairly basic memory effect, future studies examining smaller or more contextual effects will require larger and more diverse samples. Given that most VR research is still conducted in laboratory settings and measuring one person at a time, this will lead to a bottleneck at the data acquisition stage. However, as VR enters the mass market, we can expect that VR crowd studies will emerge. This would then provide researchers with access to samples the size we see in survey

research but with the added opportunity to capture biobehavioral data during message reception.

Along these lines, we also see much potential for more dynamic manipulations. By this, we mean that the current study only manipulated static billboards and the messages that were shown along a virtual drive. The next step would be introducing manipulations in which the messages are contingently administered. For instance, it would be possible to show a message if the driver previously looked at another one or to show a message for as long as needed until the driver viewed it. One could also manipulate distractions dynamically, such as via concurrent radio messages. Likewise, one could manipulate user-state variables (like having hungry participants view food billboards [90], or examine the influence of targeting more broadly. These options show the enormous potential for persuasion and nudging strategies, which are, of course, a double-edged sword: On the one hand, these could be leveraged to improve the effectiveness of health communication. On the other hand, they could be used for commercial advertising. Regardless of the intent of the messenger, however, it is undoubtedly the case that such applications would bring communicators closer to the long-standing goal of being able to "give the right message to the right recipient, at the right moment in time."

## Summary and conclusion

In sum, we suggest a VR billboard eye-tracking paradigm to study the causal path from exposure to reception and retention of messages. This paradigm allows for studying incidental memory formation across settings but with exquisite experimental control and integrated biobehavioral measurements. The result that fixations are related to memory confirms the theorized link between exposure/attention and retention/memory, underscoring the potential for this paradigm to study memory in real-world contexts and communication effects in the new information ecosystem.

## Author Contributions

**Conceptualization:** Ralf Schmälzle, Sue Lim, Gary Bente.

**Data curation:** Ralf Schmälzle, Sue Lim, Hee Jung Cho, Juncheng Wu.

**Formal analysis:** Ralf Schmälzle, Sue Lim.

**Investigation:** Ralf Schmälzle, Sue Lim.

**Methodology:** Ralf Schmälzle, Sue Lim, Hee Jung Cho, Juncheng Wu, Gary Bente.

**Project administration:** Ralf Schmälzle.

**Resources:** Ralf Schmälzle, Gary Bente.

**Software:** Ralf Schmälzle, Juncheng Wu, Gary Bente.

**Supervision:** Ralf Schmälzle.

**Validation:** Ralf Schmälzle.

**Visualization:** Ralf Schmälzle.

**Writing – original draft:** Ralf Schmälzle.

**Writing – review & editing:** Ralf Schmälzle, Sue Lim, Hee Jung Cho, Juncheng Wu, Gary Bente.

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
