## [Decision Letter · Decision Letter 0]

18 Jun 2023

PONE-D-23-04527The VR Billboard Paradigm: Using VR and Eye-tracking to Examine the Exposure-Reception-Retention Link in Realistic Communication EnvironmentsPLOS ONE

Dear Dr. Schmälzle,

Thank you for submitting your manuscript to PLOS ONE. After careful consideration, we feel that it has merit but does not fully meet PLOS ONE’s publication criteria as it currently stands. Therefore, we invite you to submit a revised version of the manuscript that addresses the points raised during the review process.

We look forward to receiving your revised manuscript.

Kind regards,

Ali B. Mahmoud, Ph.D.

Academic Editor

PLOS ONE

Journal Requirements:

2. Please provide additional details regarding ethical approval in the body of your manuscript. In the Methods section, please ensure that you have specified the name of the IRB/ethics committee that approved your study.

Reviewers' comments:

Reviewer's Responses to Questions

**Comments to the Author**

1. Is the manuscript technically sound, and do the data support the conclusions?

Reviewer #1: Yes

Reviewer #2: Yes

2. Has the statistical analysis been performed appropriately and rigorously? 

Reviewer #1: I Don't Know

Reviewer #2: Yes

3. Have the authors made all data underlying the findings in their manuscript fully available?

Reviewer #1: Yes

Reviewer #2: No

4. Is the manuscript presented in an intelligible fashion and written in standard English?

Reviewer #1: No

Reviewer #2: Yes

5. Review Comments to the Author

Reviewer #1: I like the idea of combined effort in using VR and Eye tracking. But I have a hard time understanding the manuscript. Frankly, I often second-guessed what the authors were trying to communicate. Simple English is preferred.

The literature review only talks about how great the idea of combined usage of VR and Eyetracking is. Still, there is no reason for the experimental manipulation in the methodology. It seems that manipulating “free viewing” and “counting trash” is related to cognitive load. Authors may use the Limited Capacity Model of Motivated Mediated Message Processing (LC4MP) theory for reasoning. Please also justify why the experiment deliberately had half of the billboards be “health-related topics” and others be “typical advertisements.” Does “health-related topics” mean public service ad (PSA)? And I think the experiment is trying to test something related to distance, as subjects need to provide feedback on spatial presence. But I don’t follow the meaning “the distance between successive billboards was assigned randomly and then kept fixed across participants.”

And authors should be careful in claiming the realism of VR. Not all VR looks like reality. It all depends on the production. Most VRs I have seen for experiments are poorly made due to the production cost. I can’t access the stimuli in this study. But if the authors want to justify how real it looks, it is worthwhile to elaborate more about it. For example, does the cloud in the sky (seen in the screenshot) move and vary in shape? Besides the billboard, is the surrounding environment changing with that 10mins simulation (say, another car passed by, more grass, a hut appeared somewhere)? And what’s the virtual car speed? Are the brands featured on all billboards in fictitious names?

Measurements should be clearly stated. Is the frequency of fixation or the duration of fixation being used for analysis? Or both? And it is unclear how the relationship between fixation and recall/recognition data is calculated. Is it code “1” for billboards being fixated and recalled/recognized and code “0” for others? Or is there some variation in the fixation measurement based on the number of times or duration? Also, are “spatial presence” and “occurrence of symptoms” open questions?

ANOVA is a statistical test to analyze the difference between the means of more than two groups. As far as I understand, the analysis compares two groups: free viewing vs. trash counting. Or is it indeed a multiple-group analysis? The statistical analysis should clearly state what is being compared.

The dose-response relationship is about the correlation between the effect and the volume of the same dose. Since the experiment has not repeatedly shown the same billboard, there is no dose-response relationship. It is not appropriate to claim a causal pathway of the dose-response relationship.

I would strongly advise authors to have thorough proofreading the manuscript. References 2 and 11 are missing at the back. There is no figure label at the top or bottom of the figures. Do check on the grammar. For example, “If true (see, e.g., the rebranding of the social media company FaceBook to Meta), VR might be on the way to becoming a messaging environment in and of itself.”

Reviewer #2: 1. This claim "...we know shockingly little about how many messages we encounter in our daily lives" on p.5 is inconsistent with a point you make earlier in the manuscript. A variety of data sources (e.g., Nielsen data) give us good indicators about opportunities for exposure. Can you clarify?

2. On p.5, you write "our knowledge and theories about exposure and its transmission into message effects are woefully incomplete." Is "woefully incomplete" really the current state of our knowledge? Visual attention studies are not the only way of measuring message exposure and plenty of work has been done examining exposure to and subsequent recall of auditory messages. It seems you overstate the novelty of the project. Even in the visual domain, using VR, there is work looking at exposure and recall (e.g., https://doi.org/10.1080/15252019.2020.1846642). Of course, that work has not used eye-tracking (as you have), but others have used eye tracking in increasingly naturalistic advertising contexts (e.g., work by Claire Segijn). A more thorough review of the literature, and tempering of your claims, is warranted.

3. Related, some theoretical starting points exist for thinking about the problem. For instance, Lang's LC4MP provides a decent foundation for thinking about exposure -> recall as you are interested in here. I'm not saying you have to use or cite Lang or the LC4MP. But I don't think we're at ground zero when it comes to theorizing about the issue.

4. Can you provide a citation to back up this claim on p.8 "It is well documented that such a parallel, the attention-consuming task will distract participants and should lead to fewer fixations to the billboard messages."

5. Memory research in highly controlled experimental media contexts shows moderate effect sizes for memory (https://doi.org/10.1080/23808985.2020.1839939) and more recent work shows that VR can actually dampen memory for messages, especially in instances of high spatial presence like your study (e.g., https://doi.org/10.1016/j.compedu.2022.104532). Why was a large effect size selected a priori and what is your justification? Please note, I do see that you estimate power at .95 and then round up. But how does it look with a smaller effect size?

6. For the recognition data, is there a reason you didn't calculate signal detection measures (https://doi.org/10.3758/BF03207704)? This would give more insight into the thing you are worried about (e.g., "gauge participants’ tendency to recognize all messages as seen" p.10). It would also give more insight into the psychological processes related to recognition. There is an imbalance between targets and foils (20 vs 4), but it should be possible to calculate signal detection measures, and there are some nice packages for doing this easily in R (e.g., https://rdrr.io/cran/psycho/man/dprime.html).

7. Is it possible to get an F1 score your your machine learning model?

Minor comment:

1. Citation #11 is missing from the reference list.

6. PLOS authors have the option to publish the peer review history of their article (what does this mean?). If published, this will include your full peer review and any attached files.

Reviewer #1: No

Reviewer #2: **Yes: **Richard Huskey

---

## [Author Response · Author response to Decision Letter 0]

21 Jul 2023

We have uploaded a point-by-point response as a separate document

---

## [Decision Letter · Decision Letter 1]

25 Aug 2023

PONE-D-23-04527R1The VR Billboard Paradigm: Using VR and Eye-tracking to Examine the Exposure-Reception-Retention Link in Realistic Communication EnvironmentsPLOS ONE

Dear Dr. Schmälzle,

Thank you for submitting your manuscript to PLOS ONE. After careful consideration, we feel that it has merit but does not fully meet PLOS ONE’s publication criteria as it currently stands. Therefore, we invite you to submit a revised version of the manuscript that addresses the points raised during the review process.

We look forward to receiving your revised manuscript.

Kind regards,

Ali B. Mahmoud, Ph.D.

Academic Editor

PLOS ONE

Reviewers' comments:

Reviewer's Responses to Questions

**Comments to the Author**

1. If the authors have adequately addressed your comments raised in a previous round of review and you feel that this manuscript is now acceptable for publication, you may indicate that here to bypass the “Comments to the Author” section, enter your conflict of interest statement in the “Confidential to Editor” section, and submit your "Accept" recommendation.

Reviewer #1: (No Response)

Reviewer #2: All comments have been addressed

2. Is the manuscript technically sound, and do the data support the conclusions?

Reviewer #1: Yes

Reviewer #2: Yes

3. Has the statistical analysis been performed appropriately and rigorously? 

Reviewer #1: No

Reviewer #2: Yes

4. Have the authors made all data underlying the findings in their manuscript fully available?

Reviewer #1: Yes

Reviewer #2: Yes

5. Is the manuscript presented in an intelligible fashion and written in standard English?

Reviewer #1: Yes

Reviewer #2: Yes

6. Review Comments to the Author

Reviewer #1: Thank you for providing the revised manuscript. The authors have made improvements that make it easier to understand their intended message.

After reviewing the manuscript, it appears that three main topics have emerged. However, the write-up lacks a comprehensive and coherent presentation, and it seems a bit scattered. I would suggest that the authors streamline one of the topics and provide a better narrative walk-through.

1. The combined use of VR and eyetracker: The literature review and discussion primarily focus on this topic. However, the write-up fails to establish a clear connection with the actual experiment, which is related to exposure and cognitive resources. Although the discussion covers the use of technology, it is not the main priority.

2. The importance of exposure measurement: The literature review does not provide much elucidation on this topic, and there is no theoretical framework presented. Surprisingly, this topic is introduced in the discussion section. It is unclear why the authors chose to compare fixations between the two experimental conditions instead of conducting a correlation or regression analysis to link the number of exposures to recall. Furthermore, please rephrase line 352, which states that billboards that are never looked at are practically never recalled. If subjects can recall a billboard without seeing it, it would be considered blind guessing.

3. The allocation of cognitive resources: Similar to the exposure measurement topic, there is no discussion or theoretical framework about this in the literature review section. The authors mention a 2x2 ANOVA analysis to clarify this topic, but it is unclear how the groups were divided for the fixation count. Were they divided into high and low groups? Additionally, the write-up mentions a resulting table in line 332, but I couldn't locate it.

To improve the manuscript, I recommend the authors focus on one topic and provide a clearer narrative throughout. They should also address the issues related to the importance of exposure measurement and the allocation of cognitive resources by providing a more comprehensive literature review and theoretical framework. Additionally, clarifying the analysis methods and providing the missing table would enhance the manuscript's clarity.

Reviewer #2: Thank you for such a thoughtful response letter. All my comments have been addressed. I hope to see this published soon.

Richard Huskey

7. PLOS authors have the option to publish the peer review history of their article (what does this mean?). If published, this will include your full peer review and any attached files.

Reviewer #1: No

Reviewer #2: **Yes: **Richard Huskey

---

## [Author Response · Author response to Decision Letter 1]

29 Aug 2023

A point-by-point response to the reviewer comments is attached as a separate file.

---

## [Decision Letter · Decision Letter 2]

6 Sep 2023

PONE-D-23-04527R2Examining the Exposure-Reception-Retention Link in Realistic Communication Environments via VR and Eye-tracking: The VR Billboard ParadigmPLOS ONE

Dear Dr. Schmälzle,

Thank you for submitting your manuscript to PLOS ONE. After careful consideration, we feel that it has merit but does not fully meet PLOS ONE’s publication criteria as it currently stands. Therefore, we invite you to submit a revised version of the manuscript that addresses the points raised during the review process.

We look forward to receiving your revised manuscript.

Kind regards,

Ali B. Mahmoud, Ph.D.

Academic Editor

PLOS ONE

Journal Requirements:

Reviewers' comments:

Reviewer's Responses to Questions

**Comments to the Author**

1. If the authors have adequately addressed your comments raised in a previous round of review and you feel that this manuscript is now acceptable for publication, you may indicate that here to bypass the “Comments to the Author” section, enter your conflict of interest statement in the “Confidential to Editor” section, and submit your "Accept" recommendation.

Reviewer #1: All comments have been addressed

2. Is the manuscript technically sound, and do the data support the conclusions?

Reviewer #1: Yes

3. Has the statistical analysis been performed appropriately and rigorously? 

Reviewer #1: Yes

4. Have the authors made all data underlying the findings in their manuscript fully available?

Reviewer #1: Yes

5. Is the manuscript presented in an intelligible fashion and written in standard English?

Reviewer #1: Yes

6. Review Comments to the Author

Reviewer #1: Thanks for the feedback. All looks good.

I want to clarify one aspect of the ANOVA analysis. Is the dependent variable the memory performance/recall? With the current writing, I have some doubts about whether there are other complex mathematical calculations involved in defining the two groups. If the dependent variable is indeed recall, it might be better to remove the phrase "were later recalled or recognized" in line 351 and "subsequent memory performance" in line 356. Additionally, it might be more straightforward to state that recall is the dependent variable, rather than using a blanket example.

After that, the manuscript is good to go.

7. PLOS authors have the option to publish the peer review history of their article (what does this mean?). If published, this will include your full peer review and any attached files.

Reviewer #1: No

---

## [Author Response · Author response to Decision Letter 2]

7 Sep 2023

we have attached a separate response to reviewers document

---

## [Decision Letter · Decision Letter 3]

11 Sep 2023

Examining the Exposure-Reception-Retention Link in Realistic Communication Environments via VR and Eye-tracking: The VR Billboard Paradigm

PONE-D-23-04527R3

Dear Dr. Schmälzle,

We’re pleased to inform you that your manuscript has been judged scientifically suitable for publication and will be formally accepted for publication once it meets all outstanding technical requirements.

Kind regards,

Ali B. Mahmoud, Ph.D.

Academic Editor

PLOS ONE

Additional Editor Comments (optional):

Reviewers' comments:

Reviewer's Responses to Questions

**Comments to the Author**

1. If the authors have adequately addressed your comments raised in a previous round of review and you feel that this manuscript is now acceptable for publication, you may indicate that here to bypass the “Comments to the Author” section, enter your conflict of interest statement in the “Confidential to Editor” section, and submit your "Accept" recommendation.

Reviewer #1: All comments have been addressed

2. Is the manuscript technically sound, and do the data support the conclusions?

Reviewer #1: Yes

3. Has the statistical analysis been performed appropriately and rigorously? 

Reviewer #1: Yes

4. Have the authors made all data underlying the findings in their manuscript fully available?

Reviewer #1: Yes

5. Is the manuscript presented in an intelligible fashion and written in standard English?

Reviewer #1: Yes

6. Review Comments to the Author

Reviewer #1: Thanks for all the revisions.

And great to see the advancements in VR and eyetracking research.

Look forward for the publication of the paper.

7. PLOS authors have the option to publish the peer review history of their article (what does this mean?). If published, this will include your full peer review and any attached files.

Reviewer #1: No

---

## [Editor Report · Acceptance letter]

12 Sep 2023

PONE-D-23-04527R3 

Examining the Exposure-Reception-Retention Link in Realistic Communication Environments via VR and Eye-tracking: The VR Billboard Paradigm 

Dear Dr. Schmälzle:

I'm pleased to inform you that your manuscript has been deemed suitable for publication in PLOS ONE. Congratulations! Your manuscript is now with our production department. 

Kind regards, 

on behalf of

Dr. Ali B. Mahmoud 

Academic Editor

PLOS ONE